# Point-of-care testing in a high-income country paediatric emergency department: a qualitative study in Sweden

Reza Rasti [1,2,3] Johanna Brännström,[1] Andreas Mårtensson,[4] Ingela Zenk,[5] Jesper Gantelius,[6] Giulia Gaudenzi [1,6] Helle Mölsted Alvesson,[1] Tobias Alfvén[1,5]

HMA and TA contributed equally.

For numbered affiliations see end of article.

**Correspondence to**
Dr Reza Rasti; reza.rasti@ki.se

## ABSTRACT

**Objectives** In many resource-limited health systems, point-of-care tests (POCTs) are the only means for clinical patient sample analyses. However, the speed and simplicity of POCTs also makes their use appealing to clinicians in high-income countries (HICs), despite greater laboratory accessibility. Although also part of the clinical routine in HICs, clinician perceptions of the utility of POCTs are relatively unknown in such settings as compared with others. In a Swedish paediatric emergency department (PED) where POCT use is routine, we aimed to characterise healthcare providers' perspectives on the clinical utility of POCTs and explore their implementation in the local setting; to discuss and compare such perspectives, to those reported in other settings; and finally, to gather requests for ideal novel POCTs.

**Design** Qualitative focus group discussions study. A data-driven content analysis approach was used for analysis.

**Setting** The PED of a secondary paediatric hospital in Stockholm, Sweden.

**Participants** Twenty-four healthcare providers clinically active at the PED were enrolled in six focus groups.

**Results** A range of POCTs was routinely used. The emerging theme *Utility of our POCT use is double-edged* illustrated the perceived utility of POCTs. While POCT services were considered to have clinical and social value, the local routine for their use was named to distract clinicians from the care for patients. Requests were made for ideal POCTs and their implementation.

**Conclusion** Despite their clinical integration, deficient implementation routines limit the benefits of POCT services to this well-resourced paediatric clinic. As such deficiencies are shared with other settings, it is suggested that some characteristics of POCTs and of their utility are less related to resource level and more to policy deficiency. To address this, we propose the appointment of skilled laboratory personnel as ambassadors to hospital clinics offering POCT services, to ensure higher utility of such services.

## Strengths and limitations of this study

► This is one of few studies to present the use and utility of point-of-care tests in a high-income country paediatric hospital setting, directly from the perspective of its clinical staff.

► The findings of the relatively small-sized single-centre study may be contextual, yet we believe our main conclusions to be generalisable as most of our findings are compatible with those reported in other settings.

► The study is strengthened by the diversity of its participants, corresponding to the multiprofessional staffing of the paediatric emergency department.

increasing availability of clinically graded point-of-care tests (POCTs) has brought the means for sample analyses to the point of patient care, from having previously been confined to traditional laboratory settings.[2] POCTs can be described as diagnostic tests performed near the patient or treatment facility, with a short time-to-result, that may lead to a rapid change in patient management.[3]

In many resource-limited health systems with scarce laboratory resources, POCTs for infectious diseases such as malaria, HIV and syphilis are the only means for sample analyses.[4] Experiences with and challenges of POCTs from a healthcare provider perspective have been previously reported for low-income and middle-income countries (LMICs), and at primary or adult care facilities in high-income countries (HICs).[5–10] However, to our knowledge, there is only one published report of those aspects of POCTs for paediatric emergency care facilities in HICs, despite such tests being part of routine clinical practice.[11] In such settings, POCT merits, such as short time-to-result and

## INTRODUCTION

Laboratory analyses of specific biomarkers or detection of microorganisms are a central part of the clinical diagnostic process.[1] The

simplicity, have resulted in their use, despite high accessibility to advanced laboratory facilities.

Recently, we reported the use of POCTs from the perspective of healthcare providers in a low-income country, that is, Uganda.[5] There, we identified several strengths and shortcomings of available POCTs and in the way they had been implemented locally. Despite some of these findings being potentially transferable to HICs, aspects of the use and utility of POCTs in HIC paediatric hospital settings remain only partially understood.

Thus, the literature has left several questions unanswered. These include How is the routine of point-of-care testing in a HIC paediatric emergency department (PED) setting perceived by users? How do testing experiences compare to those reported in other settings? Are there any conclusions to be drawn about how POCTs have been implemented in such a setting? What requests are there for novel POCTs?

By gathering experiences and perspectives of the use of POCTs directly from end-users in a well-resourced PED setting, this study aimed to provide a puzzle piece that promotes continued discussion on how to strengthen the role and utility of POCTs in paediatric care. Ultimately, we hope our findings can contribute to an increased utility and benefit of POCTs for healthcare providers and care seekers alike.

## METHODS
### Setting
Stockholm is the capital of Sweden, a HIC of approximately 10.4 million inhabitants.[12] The study was conducted in 2017 at a secondary paediatric hospital, housing one of three PEDs serving the greater Stockholm County, with a total population of approximately 2.4 million.[12] Along with the PED, the hospital comprises two general paediatric inpatient wards (25 beds in total), two neonatal intensive care units with two neonatal wards, several in-house paediatric outpatient units and three additional satellite out-patient units.

The PED is visited by 30 000 children aged 0–18 years annually, respiratory tract infections being the leading cause for consultation. It is staffed by paediatric consultants/specialists, residents, newly graduated junior medical doctors (filling a temporary position to qualify for clinical rotations leading to medical license), nurses and nurse assistants. Outside office hours, the PED is staffed by nurses, nurse assistants, junior medical doctors and one to two residents/specialists managing the PED as well as the inpatient wards, the neonatal units and the adjacent delivery and antenatal care units of the hospital. During these hours, paediatric and neonatal consultants are on-call, ready to support the physicians on-site by telephone. In addition to centralised laboratory and radiology functions being present at the hospital, the PED and wards at the hospital are equipped with a variety of POCTs conducted locally.

### Study design and participants
All healthcare providers (approximately 180), including nurses, nurse assistants, senior paediatricians (consultants), junior doctors and paediatric residents, clinically active at the PED were invited to participate in this study through convenient sampling, by e-mail, or via invitations at staff meetings. In total, 24 individuals agreed to participate and were placed into one of six focus groups, according to profession, to promote professional homogeneity.[13 14] The groups were composed of three to six participants (table 1) and engaged in moderated qualitative focus group discussions (FGDs) in Swedish from March to December 2017. Five of the FGDs were held in a conference room at the hospital, and one FGD (with consultants) was conducted off-site during an annual clinic gathering.

### Data collection and analysis
Based on reports of studies with similar approaches and following discussions within the study team, an interview guide (online supplemental file 1) was developed, with the main topics being: *experiences of using current point-of-care technology*; *what is most important when point-of-care tests are used*; and *what an ideal point-of-care test would include*.[5 7 15 16] The interview guide was pilot tested in the 'Junior Doctors group 1', before being used in the subsequent FGDs. As a few topics of the guide did not instigate discussion in the earlier groups, they were disregarded in the following group discussions.

All FGDs were moderated by either two or three of the authors (RR, HMA, JB), with one moderator taking an active role and the other(s) having supporting roles. Initiating each discussion, moderators presented the purpose of the study and the general forms for discussions, and defined POCTs according to Schito *et al*.[3] The FGDs were audiorecorded and transcribed verbatim in Swedish by RR and JB.[13 17 18] All transcripts were de-identified during data transcription and in the resulting manuscript.

A data-driven approach for qualitative content analysis, grounded in phenomenology and hermeneutic interpretations, was taken, as it promotes an explorative interpretivist paradigm suitable for addressing the under-explored research questions of the study.[19–21] Analysis was supported by NVivo for Mac V.11.4.3 (QSR International). Meaning units were identified and coded in English. Once all data had been coded, each code was re-evaluated and compared with the others. This process was repeated several times by RR (with previous experience in qualitative content analysis), on discussions with HMA (a medical anthropologist with vast experience in qualitative studies), resulting in the merging of matching codes and recoding of meaning units whose substances were better understood after repeated reading. The obtained set of codes was then abstracted into subcategories, categories and an overarching theme. Following continued discussions within the study team, the structure resulting from this process was revisited on several occasions until agreement of saturation had been reached.[20] The Standards for Reporting Qualitative Research checklist was followed for the manuscript writing.[22]

**Table 1** Composition of the focus groups and characteristics of the participants

| | Focus group | | | | | |
| | Nurse assistants | Nurses | Junior doctors group 1 | Junior doctors group 2 | Paediatric residents | Paediatric consultants |
| | N (=24) | | | | | |
|---|---|---|---|---|---|---|
| Group size | 5 | 3 | 3 | 4 | 3 | 6 |
| **Age (years)** | | | | | | |
| 21–30 | – | – | 1 | 3 | 1 | – |
| 31–40 | 2 | 2 | 2 | 1 | 2 | – |
| 41–50 | – | 1 | – | – | – | 2 |
| >51 | 3 | – | – | – | – | 4 |
| **Sex** | | | | | | |
| Female | 5 | 3 | 3 | 2 | 3 | 3 |
| Male | | | | 2 | | 3 |
| **Training** | | | | | | |
| **Years since graduation** | | | | | | |
| 0–3 | – | – | 3 | 4 | – | – |
| 4–7 | – | – | – | – | 2 | – |
| 8–10 | – | 1 | – | – | 1 | – |
| 11–14 | – | 1 | – | – | – | – |
| >15 | – | 1 | – | – | – | 6 |
| **Frequency of POCT use (number of tests per week)** | | | * | | | |
| 0–10 | – | 1 | | – | 1 | 4 |
| 11–20 | 1 | 2 | | 2 | – | – |
| >20 | 4 | – | | 2 | 2 | 2 |

*Data not collected for this pilot group.
POCT, point-of-care test.

## Patient and public involvement

Patients or members of the public were not involved in this study.

## RESULTS

Data analyses yielded nine subcategories that were grouped into two categories and abstracted into one theme: '*Utility of our POCT use is double-edged*' (table 2). Categories and subcategories are described in detail below. In addition to the these, online supplemental file 2 gives an overview of POCTs that participants described as available to and known by them, and presents their requests for *ideal* future POCTs, when asked to be visionary.

Median FGD duration was 74 min (range: 60–81 min).

## POCTs have clinical and social values

Participants described POCT characteristics and use as favourable to meeting clinical needs in paediatric settings, as well as serving non-clinical interests, such as fulfilling the needs of care seekers, and fulfilling the personal needs of doctors.

**Table 2** Structure of theme, categories and subcategories

| Theme | Main categories | Subcategories |
|---|---|---|
| Utility of our POCT use is double-edged | POCTs have clinical and social values | Paediatric care is favoured by the use of POCTs |
| | | Reassuring guardians and fulfilling their expectations |
| | | Fulfilling personal needs of doctors |
| | Our current testing practice distracts from care for our patients | The temptation of POCTs disrupts our clinical reasoning |
| | | We are inadequately informed about our POCTs |
| | | Non-standardised use of POCTs |
| | | POCTs are not 'all good' |
| | | POCT results cannot always be trusted |

POCTs, point-of-care tests.

**Table 3** The subcategories of 'POCTs have clinical and social values' with a sample of corresponding quotes

| Paediatric care is favoured by POCT use | Reassuring guardians and fulfilling their expectations | Fulfilling personal needs of doctors |
|---|---|---|
| It is also such that babies, or babies and children in general, can be somewhat difficult to interpret sometimes. … E.g., they can say that they have a stomachache, but that could be anything! And then it is very pleasant to have the dipstick [urinalysis POCT] just to rule out a urinary tract infection. I think many of these tests are for ruling out more serious…[conditions]. (Junior doctor group 2) | To know what's wrong, that's what I think it is, because they [guardians] want to know what's wrong. Not just that it is a virus, which virus and why, why is it like this. (Nurse assistant group) | The technology is hard to beat, you can say. Either that or grey hair; I'm waiting for either one. (Junior doctor group 2) |
| I think all of them are smooth … a CRP [C-reactive Protein] then you can be like "we have the smallest needle, the little prick … it's one drop and it doesn't hurt". Pretty easy to sell. (Nurse assistant group) | Yes, but during the peak seasons there is much talk in media about RS [Respiratory Syncytial Virus] and such, and then the parents, when they come with their child, they wonder if it is RS. So, it is also that they perhaps are sometimes pressuring. (Nurse group) | I thought about it a bit, what you were talking about when it [the POCT] is taken too much for curiosity or uncertainty of the doctor. (Nurse group) |
| I believe that it matters for the management when you receive that it is Influenza A and it is someone who thinks he/she has double vision and his/her calves hurt, and has fever and coughs, and it's Influenza A. Then I don't need to start thinking of other neurological causes, and then I can perhaps discharge the patient. But if I had received that it's not Influenza A, then I would perhaps need to admit and investigate, so to me it matters. (Consultant group) | Sometimes I think it can be purely communicative with guardians. It can be that we now have taken an infection test, and it is low; this very much looks like a viral infection combined with having had symptoms for a couple of days, and it is low, so it doesn't suggest bacterial infection. You can go home and rest and come back if there were something. (Resident group) | Let's say that you are at home, and are called by a very inexperienced, new colleague; then I think you get like, 'but maybe we should take some extra samples', because I'm not really sure of the anamnesis you get, because I don't really know, I don't know that person as much as if I had with a more experienced colleague where I would have trusted the story more and everything. I can imagine that it gets like that. (Consultant group) |

POCTs, point-of-care tests.

The subcategories of this topic are illustrated by a sample of quotes in table 3.

### Paediatric care is favoured by POCT use

POCTs were described as easy to learn and use, and as beneficial to paediatric patients (table 3). The latter was due to POCTs being less invasive (eg, capillary blood samples or transcutaneous bilirubinometer) and requiring smaller specimen volumes than those required by the central laboratory. Sample acquisitions and test analyses were described as mostly being conducted by nurses, or by nurse assistants through special delegation.

While reflecting on the influence of POCTs on patient management, participants credited them with clearing up ambiguous situations and facilitating patient assessment (table 3).

Furthermore, POCTs were described as reliable, rapid and, in some cases, multiplex. Participants described them as simplifiers and accelerators of differential diagnostics and as guides to proper management and treatment of patients (table 3). Also, their quickness was described as shortening patient time spent in the PED. Specific situations where POCTs were described as especially useful were in triage and emergency situations. The Creactive protein test was indicated to be an aid in fever case management and for evaluating treatment response in patients admitted due to infection, and the blood gas test for following the progress of patients in respiratory distress.

### Reassuring guardians and fulfilling their expectations

All focus groups witnessed patient guardians specifically requesting diagnostic tests to be undertaken, during consultations. It was perceived that some guardians consider testing to be an essential part of patient assessment. It was also suggested that some guardians bring their children to the PED only to have them undergo testing, and several participants described decisions on testing sometimes being based on such requests (table 3). At the same time, test results were considered to have a reassuring effect on worried guardians and a pedagogical role in communication between caregivers and care seekers.

### Fulfilling personal needs of doctors

In addition to being used for the benefit of patient management and fulfilling care seeker needs, POCTs were also viewed as having personal value to doctors, especially to those with little clinical experience. The credibility of assessments and clinical decisions made by junior doctors were described as strengthened by test results (table 3).

Elaborating on testing for the benefit of the doctor, test results were thought to reduce doctors' anxiety regarding mistakes, and as having an educational function, especially

**Table 4** The subcategories of 'Our current testing practice distracts from care for our patients' with a sample of corresponding quotes

| The temptation of POCTs disrupts our clinical reasoning | We are inadequately informed about our POCTs | Non-standardised use of POCTs | POCTs are not 'all good' | POCTs cannot always be trusted |
|---|---|---|---|---|
| I can sometimes feel it is unjustified if it is a very alert child who is eating, peeing, is afebrile, and feels well, and then we take samples, like 'do a CRP on that'. But they don't have a fever, they haven't anything … they have been running around for 2–3 hours. (Nurse assistant group) | No, but I don't even know how to do a proper RS [POCT for RS virus] test so that you know this is a positive RS. I haven't had an introduction or instructions on how to take it [the sample]. (Junior doctor group 2) | I think that I probably do it differently every day, that I'm not consistent myself either… *laughter*. And it can be that the week before you had a patient who became very ill and had to go to the intensive care unit – and then maybe I take that extra CRP the week after, because I remembered that case. So, it affects your decisions. (Consultant group) | If it is a child who is extremely afraid of hospitals and needles, then even a stick in the nose is painful. (Consultant group) | I like the CRP because it is reliable … However, my interpretation of it is not reliable. So, the test result feels reliable, but I don't feel reliable in my interpretation of it all the time. (Junior doctor group 2) |
| Unfortunately, yes. And again, it is probably meant to help, but it can turn into a hindrance if it doesn't show what one hopes. (Resident group) | I have no clue how much these tests cost. (Consultant group) | No, but I think that the talk goes so differently, depending on, it's probably always what is so difficult with our job. That everybody always thinks so differently, that some consultants think so differently about what you should do. (Resident group) | And then we found out the cost … Since then, I probably have never used it [POCT] for several years now! (Resident group) | But I think that it is often that I get a negative result and then you have done a new test which has been positive. So that I don't really dare to trust it. (Junior doctor group 2) |
| The clinical picture [acumen], the clinical picture, it is the clinic picture you act on. (Resident group) | … the validation of these devices, I don't always know how it has been conducted. So that, I don't know about as an individual doctor, like, how good is this device really, now that we have it. That I don't know. (Consultant group) | But I think that a lot is at an individual level… Some doctors don't want any samples at all and some want samples of practically everything. (Nurse group) | But we don't consider the staff time required, because we think that the nurses are already there. But on the other hand, they could be managing other patients if they weren't standing there working on the reagent. (Consultant group) | But it is also somewhat how the sample is taken and how much secretion you have got. (Nurse group) |

CRP, C-reactive protein; POCTs, point-of-care tests; RS, Respiratory Syncytial Virus.

for less experienced doctors who were considered most prone to prescribing tests.

## Our current testing practice distracts from care for our patients

Concurrent with observing the benefits of POCTs for work at the paediatric clinic, participants expressed concerns regarding their use, and how POCTs could complicate things. The subcategories of this topic are illustrated by a sample of quotes in table 4.

### The temptation of POCTs disrupts our clinical reasoning

The availability, rapidness and simplicity of POCTs were described as allowing for overuse and unjustified testing, rather than relying on clinical skills. Such practices were described as a burden to nurses and nurse assistants, and subjected children to unnecessary procedures. Concurrently, test results were described as less decisive for patient management than what could be observed in the clinical picture.

Contrary to descriptions of POCTs clearing up ambiguous situations, they were also viewed as sometimes resulting in increased uncertainty. In some instances, wrongful testing procedures caused incorrect results that resulted in poor decisions. In other cases, results differed from what was expected, leading to doubt of clinical assessments and instigating further investigation. These concerns were in line with others relating to the difficulty of interpreting test results and deciding on proper action.

### We are inadequately informed about our POCTs

Participants recognised personal knowledge gaps concerning how POCTs work, their accuracy and cost, and the range of tests available to them. Some also stated that they had not been taught the correct sampling techniques or analysis procedures (table 4).

There was not a given uniform method for learning about POCTs and being trained on how to use them. Learning along the way was the most frequently named learning method.

Junior doctors were described as learning from senior colleagues or nurses, and nurses and nurse assistants from peers or through a hospital online teaching platform.

### Non-standardized use of POCTs

Participants with experience at other workplaces described their current paediatric setting as being more reliant than adult clinics on POCT analyses, and of

differing testing routines between paediatric hospitals. They also described a lack of written clinical guidelines for when and why to use POCTs. This was reflected by testimonies of in-person and inter-person non-uniformity among doctors regarding using POCTs. As junior doctors had been described more likely to use POCTs, there were also descriptions of non-stringency in patient management by more senior doctors (table 4). Also, and in the absence of written guidelines, junior doctors were said to receive different instructions depending on which senior colleague they had consulted. Another more philosophical explanation given for the lack of uniformity was the practice of medicine as an art form, as described by one consultant.

### POCTs are not 'all good'

Contrary to prior descriptions of favourable characteristics of POCTs, participants also described the tests as expensive, resource-intensive, and uncomfortable for children. Cost was also stated as an inhibitor of test utility.

### POCT results cannot always be trusted

Addressing accuracy, POCTs were described as quicker, but their results less trustworthy than those of laboratory analyses. Also, sometimes POCT results were considered difficult to read, allowing for misinterpretation. POCTs identified as having low accuracy were those for Respiratory syncytial virus, urinalysis, Streptococcus group A and Mononucleosis spot.

### Requests for POCTs

At the end of each discussion, participants were asked to name existing or non-existing POCTs that they would like to be made available to them, as well as to state ideal features of POCTs with high utility (online supplemental file 2). The possibility of POCTs becoming available to the general public and used outside clinical settings was briefly addressed by the 'Consultant group'. This raised concern that it would burden healthcare providers with worried persons seeking care due to test results that they were not qualified to interpret. However, POCTs for self-use, when used in conjunction with video consultations by healthcare providers, were thought to have a role in the not-too-distant future, and that such a scenario could also benefit patients in low-income countries with lower access to health services. All four groups of medical doctors requested POCTs capable of distinguishing between bacterial and non-bacterial infection aetiology, whereas the nurse and nurse assistant groups most frequently called for migraine and cancer detection POCTs.

Relating to requested strategies for using POCTs, participants called for clinical protocols for their use, emphasising the need for value of patient management for each conducted test.

## DISCUSSION

This is, to our knowledge, one of the first studies illustrating how POCTs are perceived by end-users in a high-resourced paediatric emergency hospital setting. Here, the praxis of POCT-driven diagnostics is a normalised part of daily operations, with a range of different tests in use. Although part of the routine clinical practice, our participants perceived the utility of their POCT use as double-edged: on one hand, being beneficial to patient management in paediatric emergency care, and having reassuring value to healthcare providers and care seekers; on the other hand, being a distraction in the work at hand, with little stringency in when and why POCTs are used. Furthermore, we illustrate a value of diagnostic testing that is not strictly clinical. Finally, we give recommendations for increased quality of POCT services and present requests for ideal POCTs and their use.

Despite POCTs being considered especially beneficial to resource-limited health systems of LMICs, this study, in line with others, shows their use to be appreciated also in a well-resourced context, despite high accessibility to more advanced laboratory diagnostics and skilled personnel.[4 5 7 8 11 23–26] POCTs are often less invasive and require smaller volumes of patient specimens than central laboratory facilities, which, together with their evolving multiplexity, promotes their use in paediatric clinics.[2 27] In our study, POCTs were merited as facilitators of patient management by accelerating differential diagnostics and triage at the PED, being favourable to paediatric patients and essential to the emergency department setting of this study. Such merits have previously been described in primary care, and recently also in paediatric hospitals in the UK.[9 11]

Meanwhile, concerns were raised regarding the accuracy of POCT results, challenges in their read-outs, insecurities in their proper use, and unexpected test results. Such concerns, together with descriptions of how POCTs can increase clinician uncertainty and cause unnecessary ancillary investigations or incorrect assessments, are compatible with prior reports.[5 10 11 28 29] Concurrently, we found knowledge gaps among our clinicians regarding essential aspects of the POCTs that they are using daily. These included insights into test accuracy and cost, awareness of correct testing procedures, and an understanding of how the analyses are performed by the assays. Unexpectedly, these knowledge gaps seem to be irrespective of clinician seniority. Despite the benefit of POCTs not requiring advanced laboratory skills, it is evident that, similar to low-resource settings, the fundaments of POCTs have not been introduced to end-users, and that teaching forums also need to be conducted for clinicians in HICs, to enhance the quality of POCT services.[2 5 30]

Our findings also highlight a lack of stringency in the prescription of POCTs; we believe this to be partly explained by inadequate training routines of end-users as well as the absence of testing protocols. Furthermore, the absence of such protocols is considered to allow for what one participant described as practicing the 'art of

medicine'. As such a practice permits incoherencies in the management of patients and can be viewed as posing a risk to patients, one might also argue that clinical medicine is seldom straightforward, and that there will always be differences in its practice.[31]

As the described deficiencies in currently available POCTs and in their use have been reported in low-income, middle-income and high-income countries,[5–11 28 29 32] it can be concluded that there are flaws in the design of currently available POCTs and their utilisation that are universal and irrespective of available resources. Also, it is evident that insufficient implementation processes are not solely due to strained resources, but rather due to inadequate attention received by stakeholders. This could have negative consequences for the adoption of new technologies and creates barriers to the full utility of such devices.[33]

The use of POCTs should be viewed as part of a diagnostic *service* offered by caregivers, and hurdles to the success of such services are intertwined with the challenges faced by the health system in which they are embedded.[4]

As the accuracy of POCTs is limited by the know-how of its user, the introduction of protocols for training and certification of end-users with requirements of regular renewal of such would minimise the risk of human error, improve quality control measures, and adhere to existing quality requirements for accreditation, as stated by the International Organisation for Standardisation (ISO 22870 and 15189).[2 26 34 35] To ensure the quality of such measures, while building on recommendations by Larsson *et al*,[2] we propose that hospital laboratory units appoint 'ambassadors' to clinics offering POCT services, in HICs. Their duties should include an inventory of local POCT needs, the procurement and implementation (including staff training) of POCTs, and repeated quality assurance of the assays and in their use. Such a task needs to be complemented by the inclusion of POCTs into existing and future patient management protocols, where applicable. Being aware of the difficulties of adhering to such recommendations in LMICs, they could arguably be feasible in high-resource settings.

As Lupton contends, medical technology has a major role in healthcare delivery and is integral to the experiences of caregivers and care seekers alike.[36] Furthermore, Armstrong *et al* illustrate how the use of diagnostic instruments can have social functions, such as fulfilling clinician duties to patients,[31] and there are numerous reports on how diagnostic testing influences care seeker satisfaction.[6 9 10 27 29 30 37] Such social values are also recognised by our participants, illustrating how test results can curb the insecurities of less experienced doctors, strengthen their credibility in dialogue with senior colleagues and care seekers, help them gain clinical expertise, and increase care seeker satisfaction. At the same time, clinically unjustified testing adds to the workload of the personnel and subjects children to procedures deemed invasive enough to be uncomfortable to them.[38 39] Also, reports of over-reliance on technology as a cause of de-skilling clinicians are echoed by our findings stating that doctors would need to rely more on their clinical skills in the scenario in which POCTs are not available to them.[5 10 29 40] Evidently, it is difficult to cater to clinical and social needs of testing, while avoiding the risks and disadvantages of unwarranted testing.

Interestingly, most of the features requested by our participants regarding *ideal* POCTs are consistent with requests made by Ugandan healthcare providers.[5] Both settings requested non-invasive, cheap, quick, foolproofly and accurate tests for communicable and non-communicable diseases, with the ability to direct clinicians towards proper patient management. Regarding specific conditions for which POCTs were requested, there were contextual differences between the two settings, reflective of differing epidemiology and availability of laboratory analyses. Testimonies of substandard POCTs, and shortage of test kits and their consumables, found in the Ugandan context, were not mirrored in this study.[5]

### Strengths and limitations

Ideally, there should be four to eight participants in each focus group.[12] Although additional participants had repeatedly been invited to the FGDs, there were last minute absentees and other obstacles to enrolling more participants, mainly due to the irregular working hours of our target participants. Yet, we consider 24 to be a large enough sample, and even though larger groups could have produced additional perspectives, they could also have limited the depth of discussions. Furthermore, as we only investigated one hospital at one level of care, some of our findings are likely contextual, limiting generalisability while providing contextual insights. However, as most of our findings are described also in other settings, we believe our main conclusions to be generalisable, despite the likelihood of differing testing practices in other paediatric hospitals. Also, as this study was conducted in 2017, its findings might not be fully representative of the current reality in the setting.

The study is strengthened by the diversity of its participants, corresponding to the multi-professional staffing of the PED, and their grouping according to profession, promoting participants to speak freely. Since authors RR (paediatric resident) and JB (medical student and employed as nurse assistant) were employed at the hospital at the time of this study, their familiarity with the participants further promoted open and friendly discussions. As author HMA was previously unknown to participants, her presence, qualitative experience, and non-clinical profession (medical anthropologist) helped keep discussions on track and limited jargon.

### CONCLUSION

In a Swedish PED setting, a range of POCTs is routinely used in clinical practice. While the utility of POCT use is seen as double-edged here, it is shown to have clinical

and social value. However, deficient implementation routines limit the benefit of POCT services. As most of our findings are shared with LMICs, it is suggested that some characteristics of POCTs and of their utility are less related to resource level and more to policy deficiency. To address this, we propose the appointment of skilled laboratory personnel as ambassadors to well-resourced hospital clinics offering POCT services, to ensure higher utility of such services.

**Author affiliations**
¹Department of Global Public Health, Karolinska Institutet, Stockholm, Sweden
²Paediatric Rheumatology Unit, Karolinska University Hospital, Stockholm, Sweden
³Paediatric Immuno-psychiatry Unit, CAP Research Centre, Stockholm Healthcare Services, Stockholm, Sweden
⁴Department of Women's and Children's Health, International Maternal and Child Health, Uppsala University, Uppsala, Sweden
⁵Sachs' Children and Youth Hospital, South General Hospital, Stockholm, Sweden
⁶Division of Nanobiotechnology, Department of Protein Science, KTH Royal Institute of Technology, Science for Life Laboratory, Stockholm, Sweden

**Acknowledgements**  The authors thank all participants of the study, and to the TREND (Trial of Respiratory infections in children for ENhanced Diagnostics) study group. Our gratitude also goes to Region Stockholm (combined residency and PhD training to RR, and ALF grant 20150503), whose financial contributions made this study possible, and to Lamont Antieau for language editing.

**Contributors**  RR, JB, HMA, TA, AM, IZ and JG planned, conceptualised and designed the study and developed the interview guide. RR was the guarantor of the study with responsibility for its overall content. RR, IZ and TA invited participants. RR, JB and HMA conducted and moderated the focus group discussions and conducted data collection. RR and JB transcribed audio files and managed the data. RR analysed and interpreted the data upon discussions with HMA. The analysis results were repeatedly revisited and discussed with authors JB, HMA, TA, AM, IZ, JG and GG before being finalised. RR wrote the manuscript and its revisions, with input from all authors. The entire team of authors reviewed and approved the final manuscript.

**Funding**  Region Stockholm (grant numbers K0175-2016 and ALF-20150303).

**Competing interests**  None declared.

**Patient consent for publication**  Not applicable.

**Ethics approval**  The study was approved by the Regional Ethical Review Board in Stockholm (ref. 2016/2296-31/1). All participants provided written informed consent and were compensated with four movie vouchers each. Soft drinks and pastries were served during focus group discussions.

**Provenance and peer review**  Not commissioned; externally peer reviewed.

**Data availability statement**  Data are available upon reasonable request. Deidentified transcripts in written Swedish will be shared upon reasonable request to the corresponding author. Audio files will not be shared, as they reveal the identity of study participants and risk their guaranteed anonymity.

**ORCID iDs**
Reza Rasti http://orcid.org/0000-0001-7816-8338
Giulia Gaudenzi http://orcid.org/0000-0003-4923-6965

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
