## [Reviewer comments · BMJ Open]

ARTICLE DETAILS

TITLE (PROVISIONAL)	Point-of-care testing in a high-income country paediatric emergency department – a qualitative study in Sweden
AUTHORS	Rasti, Reza; Brännström, Johanna; Martensson, Andreas; Zenk, Ingela; Gantelius, Jesper; Gaudenzi, Giulia; Alvesson, Helle; Alfvén, Tobias

VERSION 1 – REVIEW

REVIEWER	van Hecke, Oliver Oxford
REVIEW RETURNED	08-Jul-2021

GENERAL COMMENTS	Point-of-care testing in a high-income country paediatric emergency department – a qualitative study in Sweden July 2021 This research is about a qualitative study about the use of POCTS in one secondary care hospital/PED in Stockholm. The interviews were conducted in 2017. The aim of the study was to characterise healthcare professionals' experiences and perceptions about the POCTs they currently use and how they use them. There are some interesting and new points made e.g. as a crutch to help more junior clinicians in their dealings with parents/caregivers and senior personnel. However, the manuscript needs a major overhaul. Suggestions for major improvement Some themes are not new e.g. perceptions about the ideal POCT, and have been covered elsewhere extensively in the literature. Other themes should be expanded upon e.g. the comparison between Sweden and Uganda and how POCTs are actually used. The results section is disjointed and at times difficult to read. Some potential interesting themes are missing and do not tally with the topic guide. For example, the topic guide question asks about the most important clinical decisions that POCTs help participants make. This is not covered in any of the themes. The limitations section need to be expanded e.g. why is there a delay in publishing? what are pros/cons of conducting this study only in 1 setting? Minor comments Abstract
---

Ln49 I would use the word 'value' of POCTs rather than 'utility' throughout the manuscript. Similarly, the word 'social' (value of POCTs) is perhaps not the correct choice of words.

Ln 52 Rephrase 'the local testing practice was named to distract from the care for patients'. You mean participants though that conducting POCTs distracted from clinical care?

Ln 56 Again, here too rephrase 'Requests were made for novel POCTs and their implementation'

Article Summary

Ln 27 Similar qualitative study published in BMJ Open in UK. See link <https://pubmed.ncbi.nlm.nih.gov/33972339/>

Introduction

Ln 29-38 See comment above about other qualitative work about POCTs children

Methods

24 participants were recruited out of potentially how many healthcare providers?

Although participants were of diverse work background, the sampling is convenient rather than truly purposive.

What are the implications of organising focus groups according to profession?

Is Table 1 necessary in its current format? Perhaps best to summarise this as best you can and acknowledge that you had missing data. Or perhaps explain why you had missing data in the Limitations section.

Duration of interviews (move to Results section)

Ln 49 What does 'meaning units in Swedish were identified...' mean?

Ln 54 'Fusion of codes'? i.e. grouping similar codes together to form an initial framework of categories and candidate themes

Pg 8 Ln 3 Word choice 'Abstraction'?

Ethics

Ln. 18 Move 'All transcripts were de-identified during data transcription and in the resulting manuscript' to Methods section

Results

Ln 32 It would be unusual to have a single theme in most qualitative research. Consider that your categories may in fact be themes and subthemes?

	Table 2. Clarify subcategory 'paediatric care is favoured by the use of POCTs'. Each (sub)category needs to make sense on its own as you read it e.g. POCT fulfil the needs of clinicians Table 3. Is a saturation monitor a POCT? Or rather a POC technology? Are the subheadings used meant to be categories? This is confusing as some headings do not match Table 2 themes/subthemes. Pg12 Ln 6-7 What do you mean by 'serving non-clinical interests'? Ln 14-22 Be careful. This paragraph doesn't not come across well and it very clinician centric. Are children really simply 'clients'? Many POCTs are invasive and cause pain especially in children who are more pain averse. Ln. 50 I like the term accelerators of differential diagnostics. But what do you mean by 'proper management...of patients'? Are you saying that if one doesn't use a POCT that this is substandard care? Pg 13 11-25 This is interesting findings especially in the context of Sweden where POCT are regularly used. Table 4. Suggest you pick 1-3 quotes to illustrate the main points here, or put this entire in Appendix. Pg 16 Ln 56 Sentence 'we are inadequately...' is incomplete? Pg 17 Ln 27 Rephrase ' Descriptions of non-stringency in patient management by senior doctors..' and 'practice of medicine as an art' This is not clear to me. Table 5. What is a more rapid intoxication POCT? Discussion section Pg 20 Ln 27 The authors mention 'patient flow at the PED' in Discussion, but this is not mentioned in Results section? Pg 21 Ln 28-33 'flaws in the design of currently available POCT' paragraph. Although most people would agree that current POCT used in PED or ambulatory care settings are not fit for purpose, this is not necessarily a design issue. Pg 23 Ln 6-17 it would be of interest to expand on the differences between the two settings Sweden and Uganda and where POCT test might is useful Strengths and Limitations Pg 23 Ln 37 'Most of our findings are compatible with the findings from other settings'. This needs a reference of justification statement
--	--

REVIEWER	Ellington, Laura University of Washington, Pediatrics
REVIEW RETURNED	23-Jul-2021

GENERAL COMMENTS

Stated objectives were to 1) characterize HCPs perspectives on POCTs in Swedish ED; 2) compare to other settings; 3) suggestions for ideal POCTs.

#2 objective is not addressed in methods or results. This is only addressed in discussion and therefore is not an appropriate objective of the study itself.

Abstract:

page 3, line 15: rather than “utility”, consider perception of use. This study does not address utility.

line 24: “to compare such experiences to those reported in other settings” While touched on in the discussion, this is not directly addressed in this study

Recommend elaborating on results.

Page 3, line 54: Please clarify what is meant by “local testing practice named to distract from the care for patients”

Strengths/limitations:

- Bullet point 2 is finding of study, not a strength or limitation
- Generalizability is questioned based on a single-center study in 1 health setting

Introduction

Page 5, line 5: First sentence, please clarify. Lab analyses are important but not central for patient management, especially in pediatrics. Many of the diagnostic guidelines in pediatrics do not routinely recommend lab testing for common pediatric diagnoses: pneumonia, bronchiolitis, URI, croup, gastroenteritis

POCT definition is vague. It excludes additions referenced in the article “Moreover, it was felt that the test should not require trained laboratory personnel or clinical laboratory (centrifuge, pipettes, etc) or other infrastructural (power, air conditioning, refrigeration, running water, etc) support. It was accepted, however, that there will need to be some level of training, operator interpretation, maintenance, quality control, and assurance to ensure that the test is accurate and reproducible”

page 6, line 43: Setting: “infections” is very vague. Please be more specific.

Methods:

Please define junior doctors, residents vs consultants.

How does the number of participants reflect total number who work at the PED? - what proportion were selected/participated?

Are POCTs done at bedside or sent to lab in this health setting?

Researcher characteristics and reflexivity not addressed: who are the moderators? study team? how could their involvement have influenced the research?

Please further clarify the qualitative approach (per SRQR checklist: e.g., ethnography, grounded theory, case study, phenomenology, narrative research) and guiding theory if appropriate; identifying the research paradigm (e.g., postpositivist, constructivist/interpretivist)

	page 8, line 17: reference studies used to develop interview guide page 8, 27-29: what are these citations in reference to? Not clear based on the positioning of them Results: page 9, line 33: would recommend changing wording of themes from first person plural to third person to reflect the perceptions of those who participated Table 3. POCTs available: How do these specifics add to the manuscript as a whole? They could also be more briefly described in the Settings section. Otherwise, they need more explanation. “Paediatric care is favoured by POCT use” - Recommend clearer wording. As written, the theme does not accurately reflect the description/quotes. Throughout the results, consensus and dissenting opinions are not clear. Was there an overall opinion that POCT were generally positive or negative? How were perceptions of different care providers similar/different? The illustrative quotes chosen were more reflective of doctor opinions rather than nurse participants. The illustrative quotes are not evenly distributed by provider role or subcategory: include more quotes of subcategories highlighted in discussion. Table 2 themes not in order of text. Table 4: I would recommend integrating quotes into text or have a similar table for quotes for the first subtheme. Table 5 has a lot of information. Consider focusing on the qualities of POCT rather than tests for specific diseases that raise more questions than provide answers. Are these POCT that exist, or under development, or hopes? Were these tests that were universally asked for or all desired tests recorded. It may be interesting to consider what qualities were more desired by different provider roles. Diagnostic accuracy is described as a potential downside of POCTs. Is this a perception or based on literature? This would be important to differentiate. Page 12, line 41: Quote is cut off mid-sentence Discussion: The highlights in discussion aren’t highlighted in results. Ex: Patient flow in ED and accelerating differential diagnosis. Clearly define the novelty/strengths of this study and unique contributions/themes that may differ from or are additions to the existing body of qualitative literature on POCTs. The way the discussion is currently written, the emerging themes are in alignment with previously published work. Page 20, line 56: “it is evident that fundamentals of POCTs have not been introduced to end-users..” I recommend specifying that these findings are specific to the study site but may not reflect
--	--

	POCT roll out in other settings. Are there data on how these have been implemented in other settings? Or is this a knowledge gap? page 22, line 3: “ISO 22870...” Unclear what this is in reference to, not well-defined. This is not common knowledge. page 23 line 13: “Regarding specific conditions for which POCTs were requested, there were contextual differences between the two settings, reflective of differing epidemiology and availability of laboratory analyses.” this claim is not clear page 23, line 37: Generalizability. Based on this single center relatively small study, I would be careful about generalizability claims. There are specific subcategories of themes described that are very specific to the health setting in which the study took place. Some themes may be more generalizable, but I recommend making that distinction clear. Conclusions page 24, line 17: “as most of our findings are shared with LMICs” Please reference which studies from LMICs you’re referring to The proposal for skilled lab ambassadors – a proposal for HIC only or proposing for LMIC too? Appendix: Were the POCTs discussed for the PED only? There are questions about their use in inpatient wards. The main text emphasizes responses as they pertain to the PED and inpatient ward not described. Additional References: Li E, Dewez JE, Luu Q, Emonts M, Maconochie I, Nijman R, Yeung S. Role of point-of-care tests in the management of febrile children: a qualitative study of hospital-based doctors and nurses in England. BMJ Open. 2021 May 10;11(5):e044510. doi: 10.1136/bmjopen-2020-044510. PMID: 33972339; PMCID: PMC8112413. Hardy V, Thompson M, Alto W, Keppel GA, Hornecker J, Linares A, Robitaille B, Baldwin LM. Exploring the barriers and facilitators to use of point of care tests in family medicine clinics in the United States. BMC Fam Pract. 2016 Nov 3;17(1):149. doi: 10.1186/s12875-016-0549-1. PMID: 27809865; PMCID: PMC5093922. Patel K, Suh-Lailam BB. Implementation of point-of-care testing in a pediatric healthcare setting. Crit Rev Clin Lab Sci. 2019 Jun;56(4):239-246. doi: 10.1080/10408363.2019.1590306. Epub 2019 Apr 11. PMID: 30973797.
--	--

VERSION 1 – AUTHOR RESPONSE

Response to Dr. Laura Ellington;

Stated objectives were to 1) characterize HCPs perspectives on POCTs in Swedish ED; 2) compare to other settings; 3) suggestions for ideal POCTs.

#2 objective is not addressed in methods or results. This is only addressed in discussion and therefore is not an appropriate objective of the study itself.

- ➔ Thank you for this comment. Still, we believe that since the POCT use in HIC paediatric hospital settings are understudied, a suitable way to compare such use is to first study it (as this study has), and then compare findings to those of elsewhere. This was, as stated, a Swedish study, and did not gather any data from e.g. LMICs, or other Swedish levels of care. Yet, its findings are suitable to be compared to others, and discussed in the 'Discussion' section. In the abstract we have rephrased as "to discuss such perspectives, as compared to those reported in other settings".

Abstract:

page 3, line 15: rather than "utility", consider perception of use. This study does not address utility.

- ➔ We thank you for suggesting 'utility' to be changed to 'perception'. However, our intention with this study was to study the 'perception of utility' in the form of 'utility' being understood as 'usability', or 'usefulness' of POCTs. We have now changed it into 'clinician perceptions of the utility' in the stated section. We hope the change is satisfactory.

line 24: "to compare such experiences to those reported in other settings" While touched on in the discussion, this is not directly addressed in this study

- ➔ As this study was conducted in Sweden, it is difficult to give named comparisons of experiences in a fashion other than by discussing comparisons to other literature, in the 'Discussion' section. Also, as it is formulated "to compare such experiences to those reported in other settings". The aim in the abstract is now rephrased as "to discuss such perspectives to those reported in other settings"

Recommend elaborating on results.

Page 3, line 54: Please clarify what is meant by "local testing practice named to distract from the care for patients"

- ➔ Participants of all professions described the local, often liberal, practice of point-of-care testing to be a clinical distraction. Instead of trusting your clinical acumen, and focusing on tending to patients, nurses/nurse assistants were required to run POCTs; doctors felt that the clinical picture of patients became unclear when the results of potentially unwarranted POCTs disrupted the clinical reasoning. All due to a liberal, and somewhat non-stringent use of POCTs. We chose to label these issues as 'distractions'. This is elaborated in the 'Results' and 'Discussion' sections. The abstract has been rephrased as: "While POCT services were considered to have clinical and social value, the local routine for their use was named to distract clinicians from the care for patients."

Strengths/limitations:

Bullet point 2 is finding of study, not a strength or limitation

- ➔ Removed

Generalizability is questioned based on a single-center study in 1 health setting

- ➔ Correct, 'single-center' added to bullet point 3.

Introduction

Page 5, line 5: First sentence, please clarify. Lab analyses are important but not central for patient management, especially in pediatrics. Many of the diagnostic guidelines in pediatrics do not routinely recommend lab testing for common pediatric diagnoses: pneumonia, bronchiolitis, URI, croup, gastroenteritis

- ➔ Thank you for pointing this out. The first sentence was not specific to pediatrics, but more generally. Even though clinical traditions/cultures differ between nations (and hospitals), we acknowledge that the clinical skills, or know-how of physicians are most central in patient management. However, the patient history together with the clinical examination, and (when indicated) paraclinical investigations complement each other for proper diagnostics, and in the long run patient management. But you are correct, changed to ‘.. a central part of the clinical diagnostic process’.

POCT definition is vague. It excludes additions referenced in the article “Moreover, it was felt that the test should not require trained laboratory personnel or clinical laboratory (centrifuge, pipettes, etc) or other infrastructural (power, air conditioning, refrigeration, running water, etc) support. It was accepted, however, that there will need to be some level of training, operator interpretation, maintenance, quality control, and assurance to ensure that the test is accurate and reproducible”

- ➔ There is a variety of different definitions given for point-of-care tests. In our perspective, and as we understand the definition given by the named reference, as well as of others (e.g. Drain PK et al. Evaluating Diagnostic Point-of-Care Tests in Resource-limited Settings. Lancet Infectious Diseases 2014), any diagnostic measure that is conducted somewhat near the patient, with a short turnaround time (without ‘short’ being defined), and that has a rapid and direct impact on patient management, can be defined as a point-of-care “test”. Such a definition would permit also non in-vitro instruments to be POC. In that sense, an X-ray machine could be used in a point-of-care way, and be called a POCT. Our definition is also in line with a recent pre-print by Pandey et al: “POCT was defined as an investigative or diagnostic test utilised by staff in a clinical environment, for which results are available in a short time (within 30 minutes) to aid clinical decision making in that setting (i.e. not at a later date/time). “
DOI: 10.21203/rs.3.rs-569269/v1

page 6, line 43: Setting: “infections” is very vague. Please be more specific.

- ➔ Changed to ‘respiratory tract infections’

Methods:

Please define junior doctors, residents vs consultants.

- ➔ Please see made changes. The junior medical doctors are described in the same section. Residents are licensed doctors pursuing a five-year pediatric residency program. Consultants are specialists in pediatrics, who after some years as specialists have been promoted to consultant positions at the hospital. We believe the detailing given here to be a bit redundant for the manuscript, other than the description already given for ‘junior medical doctors.’

How does the number of participants reflect total number who work at the PED? - what proportion were selected/participated?

- ➔ Rephrased as “All health care providers (approximately 180), including nurses ...”

Are POCTs done at bedside or sent to lab in this health setting?

- ➔ Changed to " the PED and wards at the hospital are equipped with a variety of POCTs conducted locally".

Researcher characteristics and reflexivity not addressed: who are the moderators? study team? how could their involvement have influenced the research?

- ➔ The moderators and their involvement are addressed in section 'Strengths and limitations'. Two additional sentences added to 'Data collection and analysis'. The study team consists of the authors who were involved with planning and executing the data collection phase of the study.

Please further clarify the qualitative approach (per SRQR checklist: e.g., ethnography, grounded theory, case study, phenomenology, narrative research) and guiding theory if appropriate; identifying the research paradigm (e.g., postpositivist, constructivist/ interpretivist)

- ➔ Rewritten as "A data-driven approach for qualitative content analysis, grounded in phenomenology and hermeneutic interpretations, was taken, as it promotes an explorative interpretive paradigm suitable for addressing the underexplored research questions of the study"

page 8, line 17: reference studies used to develop interview guide

- ➔ Corrected

page 8, 27-29: what are these citations in reference to? Not clear based on the positioning of them

- ➔ Thank you for pointing out. The references were for the studies used to develop interview guide (see previous comment). References moved.

Results:

page 9, line 33: would recommend changing wording of themes from first person plural to third person to reflect the perceptions of those who participated

- ➔ Main theme re-labelled as "Utility of our POCT use is double-edged". We believe the subjectivity of the participants are well illuminated by the existing wording of the (sub)categories. E.g. "Our current testing practice...", and "We are inadequately informed ..."

Table 3. POCTs available: How do these specifics add to the manuscript as a whole? They could also be more briefly described in the Settings section. Otherwise, they need more explanation.

- ➔ You raise a good question. The stated POCTs are not necessarily all POCTs actually available to participants, but merely the ones mentioned by them. However, they cover almost all tests that were available at the PED at that time. We discussed whether this table should be omitted, yet decided to keep it but move it to Appendix 2. The rationale for keeping it was that it gives a good example of the range of POCTs in daily use in this setting, and can be compared to what we found in our previous study in Uganda (reference 5 - Rasti et al, PLOS ONE, 2017)

"Paediatric care is favoured by POCT use" - Recommend clearer wording. As written, the theme does not accurately reflect the description/quotes.

- ➔ The paediatric word of this sentence/category is to be stressed. POCTs are especially valuable in paediatrics, as they are less invasive and require less volume. Also, the patient

care, or management in general, is favoured as POCT results “clear up amgiguous situations”. Hence the chosing of “Paediatric care is favoured by POCT use”.

Throughout the results, consensus and dissenting opinions are not clear. Was there an overall opinion that POCT were generally positive or negative?

How were perceptions of different care providers similar/different? The illustrative quotes chosen were more reflective of doctor opinions rather than nurse participants.

- The overall opinion was that their use was double-edged. It was underlined that POCTs had benefits in this paediatric setting, for the sake of patients, as well as for guardians and care-givers. However, there were negative issues, especially related to how/when/why they were used, that were described by all groups. So, a participant that considered POCTs as positive and appreciated having them available, could also show dissenting opinion on their performance, as well as their use, by themselves as well as of their colleagues. There were. e.g. nurses and nurse assistants appreciated the less-invasiveness of POCTs, as compared to sampling for laboratory analyses, and their speed. They liked how guardians could be comforted by test results. At the same time, they considered that the prescribing doctors were burdening them with unjustified testing, which would also subjecte children to unnecessary discomfort. In all, there was no clear distinction in that certain professions were positive about POCTs, while other professions being negative. All groups showed dissenting feelings.

The illustrative quotes are not evenly distributed by provider role or subcategory: include more quotes of subcategories highlighted in discussion.

- Even though it is recommended to avoid quantification in qualitative studies, 2/3 of participants were doctors, and it would be natural for the manuscript to contain more doctors’ quotes as there are more of them in the transcripts. However, we have not had an intention of presenting quotes according to profession proportion. We have chosen the most illustrative quotes, irrespective of their source. Some doctors’ quotes have now been replaced with quotes from nurses and nurse assistants. Also, each subcategory is now represented by three quotes (in Table 3 and 4).

Table 2 themes not in order of text.

- Table 2 updated.

Table 4: I would recommend integrating quotes into text or have a similar table for quotes for the first subtheme.

- Thank you for the suggestion. A similar table has been created for the first main category. The quotes have been moved there (table 3).

Table 5 has a lot of information. Consider focusing on the qualities of POCT rather than tests for specific diseases that raise more questions than provide answers. Are these POCT that exist, or under development, or hopes? Were these tests that were universally asked for or all desired tests recorded. It may be interesting to consider what qualities were more desired by different provider roles.

- Table 5 is moved to Appendix 2. The groups were asked to give visionary requests, for specific conditions as well as for their use. The presented tests/characteristics of tests are all that were recorded. The following is added to the ‘Requests for POCT’s section: “All four groups of medical doctors requested POCTs capable of distinguishing between bacterial and non-bacterial infection aetiology, whereas the nurse and nurse assistant groups most frequently called for migraine and cancer detection POCTs.”

Diagnostic accuracy is described as a potential downside of POCTs. Is this a perception or based on literature? This would be important to differentiate.

- ➔ Are you referring to the Results section? Everything in the Results section are the perceptions of the participants.

Page 12, line 41: Quote is cut off mid-sentence

- ➔ Yes, the participant was cut off by another participant, so that he/she did not finish the sentence. But it was understood as “more serious conditions”. Have added ‘conditions’ in brackets, to the quote.

Discussion:

The highlights in discussion aren’t highlighted in results. Ex: Patient flow in ED and accelerating differential diagnosis.

- ➔ “Patient flow” has been changed to “triage”. ‘Accelerating differential diagnostics’ is concluded from the descriptions of POCTs being rapid and “simplifiers and accelerators of differential diagnostics”, as described in the Results.

Clearly define the novelty/strengths of this study and unique contributions/themes that may differ from or are additions to the existing body of qualitative literature on POCTs. The way the discussion is currently written, the emerging themes are in alignment with previously published work.

- ➔ Please see the reply to Dr van Hecke regarding this. We acknowledge our findings to be in line with existing literature. Yet, the idea for this study came after having conducted our study in Uganda. Findings from Uganda could be recognized also from the Swedish clinical paediatric setting in which the paediatrician authors of this manuscript worked. Yet, when we searched the literature, we couldn’t identify literature from HIC paediatric hospital settings. We believe this study to confirm there to be several, near identical, similarities between the two settings, despite the otherwise greatly different realities, in which they operate. This has also been elaborated on in the Discussion. As such, we believe the labelling of themes or categories, in this aspect, to be secondary.

Page 20, line 56: “it is evident that fundamentals of POCTs have not been introduced to end- users..” I recommend specifying that these findings are specific to the study site but may not reflect POCT roll out in other settings. Are there data on how these have been implemented in other settings? Or is this a knowledge gap?

- ➔ Now rephrased to be understood as specific to this study, but also existing in other settings (with references).

page 22, line 3: “ISO 22870...” Unclear what this is in reference to, not well-defined. This is not common knowledge.

- ➔ ISO = International Organization for Standardization. An international NGO promoting standardization, and accreditation of such. If not mistaken, more recognized outside the USA. This sentence has been rephrased. ISO 22870 and 15189 are specific requirements for POCTs and medical laboratories. Please see www.iso.org

page 23 line 13: “Regarding specific conditions for which POCTs were requested, there were contextual differences between the two settings, reflective of differing epidemiology and availability of laboratory analyses.” this claim is not clear

- ➔ The two settings are this Swedish one, and the Ugandan one (as described in reference Rasti, Nanjebe et al. PLOS ONE 2017). The differing epidemiology reflect the different tests requested by participants. E.g., in the Ugandan study, requests were made for POCTs/RDTs for HIV, Ebola, Tuberculosis, and others. Conditions that are quite rare in the Swedish context, and not mentioned by the Swedish participants.

page 23, line 37: Generalizability. Based on this single center relatively small study, I would be careful about generalizability claims. There are specific subcategories of themes described that are very specific to the health setting in which the study took place. Some themes may be more generalizable, but I recommend making that distinction clear.

- ➔ Please see changes made. We believe the main conclusions to be less contextual, even if there are specific categories and such that are specific to this setting.

Conclusions

page 24, line 17: "as most of our findings are shared with LMICs" Please reference which studies from LMICs you're referring to

The proposal for skilled lab ambassadors – a proposal for HIC only or proposing for LMIC too?

- ➔ Regarding references, there are plenty given in the 'Discussion'. The 'Conclusion' section does not necessarily need to have a repetition of references, as it is merely a summary of the rest of the manuscript.
Regarding the ambassadors, this is a proposal for HIC hospitals containing laboratories with skilled lab personnel, who could be used. When it comes to LMICs, there is an ongoing discussion on the best way to ensure quality of POCT services offered by caregivers on different levels. Here, it is often suggested the establishment of national/regional referral laboratories, who should be put in charge of making sure that POCT services given in the area are of high(er) quality.
'HIC' added to: "To address this, we propose the appointment of skilled laboratory personnel as ambassadors to HIC hospital clinics offering POCT services, to ensure higher utility of such services."

Appendix:

Were the POCTs discussed for the PED only? There are questions about their use in inpatient wards. The main text emphasizes responses as they pertain to the PED and inpatient ward not described.

- ➔ Our intention from the beginning was to also discuss the ward use of POCTs. However, as the discussions folded out, the focus came to be on the PED. So, the interview guide contains ward questions, but the discussions turned out differently, without digging into wards. Also, the participants were from the PED.

Additional References:

Li E, Dewez JE, Luu Q, Emonts M, Maconochie I, Nijman R, Yeung S. Role of point-of-care tests in the management of febrile children: a qualitative study of hospital-based doctors and nurses in England. *BMJ Open*. 2021 May 10;11(5):e044510. doi: 10.1136/bmjopen-2020-044510. PMID: 33972339; PMCID: PMC8112413.

- ➔ Noted and added

Hardy V, Thompson M, Alto W, Keppel GA, Hornecker J, Linares A, Robitaille B, Baldwin LM. Exploring the barriers and facilitators to use of point of care tests in family medicine clinics in the United States. *BMC Fam Pract*. 2016 Nov 3;17(1):149. doi: 10.1186/s12875-016-0549-1. PMID: 27809865; PMCID: PMC5093922.

- ➔ This study was conducted in a family medicine setting. We believe we have a good set of references, including from primary care.

Patel K, Suh-Lailam BB. Implementation of point-of-care testing in a pediatric healthcare setting. *Crit Rev Clin Lab Sci*. 2019 Jun;56(4):239-246. doi: 10.1080/10408363.2019.1590306. Epub 2019 Apr 11. PMID: 30973797.

- ➔ This reference is already cited in the manuscript.

Response to Dr van Hecke

Dr. Oliver van Hecke, Oxford

Comments to the Author:

Point-of-care testing in a high-income country paediatric emergency department – a qualitative study in Sweden

July 2021

This research is about a qualitative study about the use of POCTS in one secondary care hospital/PED in Stockholm. The interviews were conducted in 2017. The aim of the study was to characterise healthcare professionals' experiences and perceptions about the POCTs they currently use and how they use them.

There are some interesting and new points made e.g. as a crutch to help more junior clinicians in their dealings with parents/caregivers and senior personnel. However, the manuscript needs a major overhaul.

Suggestions for major improvement

Some themes are not new e.g. perceptions about the ideal POCT, and have been covered elsewhere extensively in the literature. Other themes should be expanded upon e.g. the comparison between Sweden and Uganda and how POCTs are actually used.

- ➔ We acknowledge that some results have been found elsewhere. As paediatric POCT use in high-income countries have not been as extensively studied as in low-/middle-income countries, or in adult medicine, we would like to argue that the similarities between our findings are a discovery in themselves. We have addressed this in the 'Discussion' section. Regarding the comparison to Uganda, please see reply below.

The results section is disjointed and at times difficult to read.

- ➔ We hope that the made changes have improved reading of the Results section.

Some potential interesting themes are missing and do not tally with the topic guide. For example, the topic guide question asks about the most important clinical decisions that POCTs help participants make. This is not covered in any of the themes.

- ➔ We agree that the presented results do not fully correspond to the interview guide. The reason for this is that we tried to have as much of an inductive approach as possible, without losing the bearings of the aims of the study. Even though we had additional topics (such as the one you name) to discuss with our focus group, we found that the first few groups did not “bait” on those related questions. They did not instigate any meaningful discussion, whereas the other topics did. To avoid differences in topics covered by the different groups, for the succeeding groups we intentionally put the focus on what had been discussed in prior groups. Hence, a few topics/questions received less attention. An explanation for this has now been added to the ‘Data collection and analysis’ part of ‘Methods’ section.

The limitations section need to be expanded e.g. why is there a delay in publishing? what are pros/cons of conducting this study only in 1 setting?

- ➔ Regarding the publishing delay, this was mainly due to personal reasons of the main author; since the study, he has married, became a father twice, had to re-prioritize on finishing his pediatric specialist training and advancing his PhD studies (including conducting other studies and participating in course work). At the same time, the COVID-19 pandemic limited non-clinical time for the main author as well as for the clinically active co-authors. However, the Li et al study in BMJ Open was conducted in 2018 and published in 2021.

Regarding the one-center setting of the study, this has been elaborated on in the Strengths/limitations section.

Minor comments

Abstract

Ln49 I would use the word ‘value’ of POCTs rather than ‘utility’ throughout the manuscript. Similarly, the word ‘social’ (value of POCTs) is perhaps not the correct choice of words.

- ➔ Thank you for the suggestion of using ‘value’ instead of ‘utility’. Our intention of using ‘utility’, was for it to be understood as ‘usability’, or ‘usefulness’. ‘Utility’ is also the word used by Li et al in the BMJ Open study.
Regarding ‘social’, we very much think this is a suitable word for other-than-clinical use, as suggested by reference Armstrong et al (please see ‘Discussion section’).

Ln 52 Rephrase ‘the local testing practice was named to distract from the care for patients’. You mean participants though that conducting POCTs distracted from clinical care?

- ➔ Participants of all professions described the local, often liberal, practice of point-of-care testing to be a clinical distraction. Instead of tending to the patients, nurses/nurse assistants were required to run POCTs; doctors felt that the clinical picture of patients became unclear when the results of potentially unwarranted POCTs disrupted the clinical reasoning. All due to a liberal, and somewhat non-stringent use of POCTs. We chose to label these issues as ‘distractions. Now rephrased as:
“While POCT services were considered to have clinical and social value, the local routine for their use was named to distract clinicians from the care for patients”

Ln 56 Again, here too rephrase ‘Requests were made for novel POCTs and their implementation’

- ➔ Changed to “... made for ideal POCTs ...”.

Article Summary

Ln 27 Similar qualitative study published in BMJ Open in UK. See

link <https://eur01.safelinks.protection.outlook.com/?url=https%3A%2F%2Fpubmed.ncbi.nlm.nih.gov%2F33972339%2F&data=04%7C01%7Ccreza.rasti%40ki.se%7C94ed651aef6405b3c7908d95c0aaf01%7Cbff7eef1cf4b4f32be3da1dda043c05d%7C0%7C0%7C637642023365694521%7CUnknown%7CTWFpbGZsb3d8eyJWljoIMC4wLjAwMDAiLCJQIjoiV2luMzliLCJBTiI6IjEhaWwiLCJXVCi6Mn0%3D%7C1000&sd=0&sdata=MjqeenuorwRUjQjcwlgFVlwBozgFp1eTJmm5QFvCbY%3D&reserved=0>

- ➔ Changed to "This is one of the first studies ...". Even though Li, Dewez, Luu et al study was published before this study, it was conducted in 2018 as compared to 2017 for this study. We became aware of the other study a few days after having submitted our manuscript. The other study is now cited [11].

Introduction

Ln 29-38 See comment above about other qualitative work about POCTs children
Thank you for the reference. We were made aware of the article a few days after having submitted our manuscript, as it was published shortly before. Please see previous reply.

Methods

24 participants were recruited out of potentially how many healthcare providers?

- ➔ Out of 180. Number added to the manuscript.
Although participants were of diverse work background, the sampling is convenient rather than truly purposive.
- ➔ Corrected, changed to convenient sampling.

What are the implications of organising focus groups according to profession?

- ➔ This is addressed in the Methods section and in the Strengths and Limitations sections. This was to promote professional homogeneity and encourage friendly and open discussions. We believe this approach was more suitable in this hierarchical hospital setting with a clear "chain of command", and that a more naturally occurring group structure would have discouraged participants from taking part or speaking freely, even though such a group structure would have been very interesting to follow. Please also see reference Kitzinger.

Is Table 1 necessary in its current format? Perhaps best to summarise this as best you can and acknowledge that you had missing data. Or perhaps explain why you had missing data in the Limitations section.

- ➔ Explanation added to the Table 1 footnote. The Junior doctors group 1 was also the pilot group (see 'Data collection and analysis'). The decision to collect this data was made pursuing the pilot. We believe it will be difficult to show the demographics of participants in any other way.

Duration of interviews (move to Results section)

- ➔ Done.

Ln 49 What does 'meaning units in Swedish were identified...' mean?

- ➔ 'Swedish' removed from the sentence. What we meant was that since the transcripts were in Swedish, the meaning units were also in Swedish. The translation to English was made when

those meaning units were coded. So, the codes were in English.

Ln 54 'Fusion of codes'? i.e. grouping similar codes together to form an initial framework of categories and candidate themes

- The first set of codes was quite large, as the transcripts were coded along the way. So once all the transcripts had been coded, each code was revisited. Then, we identified codes which had been labelled differently in different sections/transcripts, but once revisited became obvious that they were the same, and so these were merged, or fused. Changed to "merging of codes".

Pg 8 Ln 3 Word choice 'Abstraction'?

- Changed to "the structure resulting from this process"

Ethics

Ln. 18 Move 'All transcripts were de-identified during data transcription and in the resulting manuscript' to Methods section

- Done

Results

Ln 32 It would be unusual to have a single theme in most qualitative research. Consider that your categories may in fact be themes and subthemes?

- We agree, even though most qualitative researchers aim to be able to summarize all categories in as few themes as possible. In this case, we believe the theme 'Utility of our POCT use is double-edged' to encompass our findings.

Table 2. Clarify subcategory 'paediatric care is favoured by the use of POCTs'. Each (sub)category needs to make sense on its own as you read it e.g. POCT fulfil the needs of clinicians

- The paediatric word of this sentence/category is to be stressed. POCTs are especially valuable in paediatrics, as they are less invasive and require less volume. Also, the patient care, or management in general, is favoured as POCT results "clear up ambiguous situations". Hence the choosing of "Paediatric care is favoured by POCT use".

Table 3. Is a saturation monitor a POCT? Or rather a POC technology?

- Depends on the underlying understanding of how POCTs are defined. Please see reply to Dr Ellington on the same issue. For our focus groups, we defined POCTs to participants according to our reply to Dr Ellington, and it was on the basis of such that the discussions took place.

Are the subheadings used meant to be categories? This is confusing as some headings do not match Table 2 themes/subthemes.

- The headings in bold are main categories. Their subheadings (not bold) are subcategories.

Pg12 Ln 6-7 What do you mean by 'serving non-clinical interests'?

- Rephrased to "as well as serving non-clinical interests, such as fulfilling the needs of care seekers, and fulfilling the personal needs of doctors.

Ln 14-22 Be careful. This paragraph doesn't not come across well and it very clinician centric. Are children really simply 'clients'? Many P

OCTs are invasive and cause pain especially in children who are more pain averse.

- The Swedish word for 'clientele' (klientel) seems to have a somewhat different meaning. It is not restricted to clients/customers, but more so to "type" of individuals, in this case children. Changed to 'patients' to avoid misunderstanding. We acknowledge that even POCTs cause pain and fear in children, as described also by our participants.

Ln. 50 I like the term accelerators of differential diagnostics. But what do you mean by 'proper management...of patients'? Are you saying that if one doesn't use a POCT that this is substandard care?

- Thank you. No, that was not our intention.. We say that the participants expressed that paraclinical examinations, in this case POCTs, guide them towards choosing suitable actions for how to manage their patients. So, they [POCTs] guide the management.

Pg 13 11-25 This is interesting findings especially in the context of Sweden where POCT are regularly used.

- Thank you!

Table 4. Suggest you pick 1-3 quotes to illustrate the main points here, or put this entire in Appendix.

- Please see reply to Dr Ellington. As per her suggestion, we have now created another table 3 to contain quotes of the main category 'POCTs have clinical and social values'.

Pg 16 Ln 56 Sentence 'we are inadequately...' is incomplete?

- This is a sub-heading, and formatted according to BMJ rules.

Pg 17 Ln 27 Rephrase ' Descriptions of non-stringency in patient management by senior doctors..' and 'practice of medicine as an art' This is not clear to me.

- Changed to "... more senior doctors."
- Medicine as an 'art' is described in the Results and Discussion sections. Changed to "the practice of medicine as an art form, as described by one consultant."

Table 5. What is a more rapid intoxication POCT?

- Changed to 'Rapid urine toxicology POCT', as the toxicology tests that were available were felt to have a long turnaround time. Nb. that this table has been moved to Appendix 2.

Discussion section

Pg 20 Ln 27 The authors mention 'patient flow at the PED' in Discussion, but this is not mentioned in Results section?

- 'Patient flow' changed to 'triage', corresponding to the Results section.

Pg 21 Ln 28-33 'flaws in the design of currently available POCT' paragraph. Although most people

would agree that current POCT used in PED or ambulatory care settings are not fit for purpose, this is not necessarily a design issue.

- ➔ We agree. However, there are also design issues with many POCTs, that create unwanted situations. One example is how unclear lines in the read-out of lateral flow assays can cause confusion.

Pg 23 Ln 6-17 it would be of interest to expand on the differences between the two settings Sweden and Uganda and where POCT test might be useful

- ➔ A few additional differences added to the final paragraph of the Discussion section. However, in the Ugandan study, there were participants from different levels of the health care system, allowing discussion on differences in POCT use and need, between them. In this study all participants were from the same level of care.

Strengths and Limitations

Pg 23 Ln 37 'Most of our findings are compatible with the findings from other settings'. This needs a reference or justification statement

- ➔ This has already been addressed, together with references, in the 'Discussion' section. Here, it is only being reiterated.

Additional comments from the authors:

The former headings 'POCTs available and unavailable to us, and How we are taught about POCTs and who among us do the testing' were considered as contextual information, and not part of the content analysis and categorization. After some consideration, 'POCTs available and unavailable to us' has been moved to Appendix 2.

'How we are taught about POCTs' has been incorporated in the subcategory 'We are inadequately informed about our POCTs.'

'Who among us do the testing' has been incorporated in the subcategory 'Paediatric care is favoured by POCT use'.

Tables 3 and 5 have been moved to Appendix 2.